# IMPROVING TIME COMPLEXITY OF SPARSIFICATION ALGORITHMS

## ABSTRACT

We improve time complexity of spectral sparsification algorithms, such as Batson, Spielman and Srivastava (BSS-2009), used for iteratively computing spectral sparsifiers of $n$-vertex graphs or, more generally, for sparsifying a sum of rank-one $n \times n$ matrices, or dual-set sparsification, Boutsidis, Drineas, and Magdon-Ismail (2011) used for joint column selection. We demonstrate that for such algorithms the computations relying on matrix inversion are iterations dependent, namely inversion of large matrices at the k-th iteration can be performed using $k \times k$ matrix inversion or, for greater stability, by inverting only the lower part of a Cholesky decomposition. This improves the computational complexity of such algorithms. We propose heuristics relying on restarted sparsification taking full advantage of inverting small matrices while ensuring control on barriers as in the original algorithms. Such heuristics present an empirical interest that is validated with various numerical experiments.

## 1 INTRODUCTION

A spectral sparsifier is a reweighted sparse subgraph that approximately preserve Laplacian quadratic form. Formally, for an $n$-vertex undirected and weighted graph $G$, a subgraph $G'$ of $G$, with proper reweighting of edges is called a $(1 + \epsilon)$-spectral sparsifier if $(1 - \epsilon)\boldsymbol{L}_G \preceq \boldsymbol{L}_{G'} \preceq (1 + \epsilon)\boldsymbol{L}_G$, i.e.

$$(1 - \epsilon)\boldsymbol{x}^\top \boldsymbol{L}_G \boldsymbol{x} \leq \boldsymbol{x}^\top \boldsymbol{L}_{G'} \boldsymbol{x} \leq (1 + \epsilon)\boldsymbol{x}^\top \boldsymbol{L}_G \boldsymbol{x}, \qquad \forall \boldsymbol{x} \in \mathbb{R}^n$$

The utility of graph sparsification lies in approximating dense graphs with sparse ones, while ensuring approximation of the Laplacian. This approximation guarantees that many global properties—such as effective resistance, commute times, and cut capacities-are preserved to within provable bounds. Spectral properties can be leveraged for graph clustering; for efficiently solving min-cut/max-flow problems; for interpolating functions over the nodes. We refer to Satuluri et al. (2011); Ahn et al. (2012); Fung et al. (2011); Bravo Hermsdorff & Gunderson (2019) and references their-in.

There has been an extensive research interest for this problem, initiated by Spielman et al. Spielman & Teng (2004); Spielman & Srivastava (2008); Spielman & Teng (2011); Batson et al. (2009). In Batson et al. (2009), the first algorithm for constructing a spectral sparsifier with $n/\epsilon^2$ edges, which is optimal up to a constant (see e.g. Andoni et al. (2016)), was given. More generally, given an $n \times n$ symmetric positive semi-definite matrix $\boldsymbol{A} = \sum_{i=1}^m \boldsymbol{v}_i \boldsymbol{v}_i^\top$, the paper establishes the existence, via a deterministic constructive approach, of scalars $t_i \geq 0$ s.t. $|\{i : t_i > 0| \leq n/\epsilon^2$ and $\boldsymbol{A}' = \sum_{i=1}^m t_i \boldsymbol{v}_i \boldsymbol{v}_i^\top$ satisfies $(1 - \epsilon)\boldsymbol{A} \preceq \boldsymbol{A}' \preceq (1 + \epsilon)\boldsymbol{A}$. This result is a major theoretical breakthrough. It has shown that log factor showing in random sampling can be removed. It has also an inherent computational interest as it relies on straightforward techniques that can easily be adapted and generalized to other contexts.

The *Batson-Spielman-Srivastava (BSS)* sparse representation theorem/algorithm has become a foundational tool across theoretical computer science, applied mathematics, and machine learning. Numerous researchers have extended and applied the BSS framework in settings where linear-sized sparse representations are required—either to establish complexity-theoretic bounds or to design scalable algorithms. These techniques are particularly impactful in high-dimensional data analysis, where storing or processing the full dataset is computationally prohibitive, yet preserving the underlying geometric structure is crucial. We provide a concise overview of these developments.

**Column subset selection or sketching** The problem of selecting a representative subset of columns from a matrix—commonly referred to as *column subset selection*—has been extensively studied in

numerical linear algebra and machine learning. These techniques are essential for matrix approximation, feature selection, and data reduction in high-dimensional settings. We refer to Avron & Boutsidis (2013); Boutsidis et al. (2014) and references their-in for an overview of these problems.

A prominent line of work in column subset selection builds upon the BSS sparsification framework. Boutsidis et al. Boutsidis et al. (2011; 2013); Boutsidis & Magdon-Ismail (2013); Avron & Boutsidis (2013); Boutsidis et al. (2014) employed BSS-based techniques as subroutines in feature extraction and low-rank reconstruction. In Boutsidis et al. (2011), one of the earliest applications of BSS algorithm (called *single-set* sparsification) to deterministic and randomized feature extraction is given. Later, Boutsidis et al. (2014) introduced algorithmic variants called *dual-set spectral sparsification* (Boutsidis et al., 2014, Lemma 13) and *dual-set Frobenius sparsification* (Boutsidis et al., 2014, Lemma 14), which leverage BSS ideas to achieve asymptotically optimal column-based reconstructions under both the spectral and Frobenius norms. These algorithms rely on two key innovations: (i) fast, approximate SVD-like decompositions that estimate the dominant singular subspace without computing the full SVD; (ii) deterministic greedy selection of some of the left and right singular vectors via the dual-set variants algorithms.

Paul et al. Paul & Drineas (2016); Paul et al. (2016), employed single-set spectral sparsification for deterministic feature selection in supervised learning. Namely, it was applied as preprocessing step for regularized least-squares classification (RLSC) Paul & Drineas (2016) and to linear support vector machines (SVMs) Paul et al. (2016). It is shown that solving these problems in the reduced feature space yields approximate classifiers that are as good as the classifiers obtained using the full feature space. In particular, solving in a reduced feature space of size $\mathcal{O}(s)$, where $s$ is the number of support vectors, yields classifiers with decision boundaries and margins comparable to those obtained in the full feature space.

Spectral/Frobenius sparsification is a complementary approach for dimensionality reduction in many others contexts Cohen et al. (2015). Its performance is competitive with state-of-the-art randomized numerical linear algebra techniques Mahoney et al. (2011); Kannan & Vempala (2017), which offer practical trade-offs between efficiency and accuracy, and have been successfully applied in these contexts.

**Covariance Estimation and Sample Sparsification.** Another early application of BSS algorithm appears in the context of covariance estimation. Srivastava and Vershynin Srivastava & Vershynin (2013), apply the method to obtain bounds on the convergence of the sample covariance matrix to the true covariance matrix of a high-dimensional distribution. Their results provide deterministic bounds that are especially useful in settings where traditional concentration inequalities are insufficient due to dimensionality or sample size constraints. A related development is the work of Charikar et al. Charikar et al. (2017), who use the algorithm for sample sparsification. In this context, the goal is to sparsify a set of input vectors (samples) while preserving certain properties, such as their covariance or total variance.

**Sample Sparsification for curse fitting:** sparsification is often studied in the context of *sample selection*, that is, selecting a subset of *rows* from the data matrix. This is particularly relevant for regression problems, where the objective is to find small, informative subsets of the data samples, observe associated labels/function evaluations, and produce accurate, unbiased estimators of the full solution. Spectral sparsification techniques Batson et al. (2009); Lee & Sun (2018) were tailored to this context for establishing the existence and producing linear-sized samples w.r.t projection space dimension ensuring quasi-optimal error guarantee in expectation. We refer to Chen & Price (2019) and Dolbeault & Chkifa (2024) and references their-in for some of these results. We also refer also to Boutsidis et al. (2013); Huang et al. (2020) for the slightly similar problem of *coresets* construction, where the use of dual-set sparsification framework introduced in Boutsidis et al. (2014) is significant.

**Graphs in Machine Learning** Graph sparsification belongs to a broader class of graph reduction techniques, which include *graph sampling/coarsening/sketching/streaming/distillation* to name a few. The common concept of all these techniques is to reduce the size or complexity of a graph while approximately preserving key structural properties. They have become imperative for scaling algorithms to handle the massive graphs that commonly arise in machine learning applications. We refer to Ahn et al. (2012); Jin et al. (2022); Joly & Keriven (2024) and reference their-in for some examples

Graph sparsification techniques form an essential toolkit for processing and learning from large-scale graphs. They enable efficiency, preserve theoretical guarantees, and support a wide range of applications. They are now integrated into many modern machine learning pipelines, including: Gaussian Graphical Model Cheng et al. (2015), Fast Graph Attention Networks (GATs) Srinivasa et al. (2020), Graph Convolutional Networks (GCNs) Ahmad et al. (2021), Sparse Graph Attention Networks (SGAT) Ye & Ji (2021), Neural Networks Pruning Laenen (2023), Graph Clustering Chen et al. (2016); Chakeri et al. (2016); Sun & Zanetti (2019), Graph Learning and Laplacian Regularization Sadhanala et al. (2016); Calandriello et al. (2018), Differential Privacy Arora & Upadhyay (2019). We refer also to Dwaraknath et al. (2023) and references their-in for

**Optimization and geometry**    Sparsification also arises in *Volumetric Spanners* constructions: the work of Hazan et al. Hazan & Karnin (2016) makes use of identities similar to those found in the BSS algorithm to construct compact representations of data. In particular, Hazan uses the BSS algorithm to sparsify John's decomposition of at set of $m$ vectors in $\mathbb{R}^n$ transformed into John's position, thereby producing volumetric spanner for these vectors of order at most $12n$ and that can be constructed in $poly(m, n)$ time. We also mention Bhaskara et al. Bhaskara et al. (2023) for an overview on volumetric spanners applications and comparison with Hazan & Karnin (2016).

**Our contribution:**    BSS algorithm and the techniques based on it are all initialized with matrices $\boldsymbol{A} = \boldsymbol{0}_{n \times n}$ that are iteratively rank-one updated. They all require $n \times n$ matrix inversions at every iteration (or phase Lee & Sun (2018)). These matrices have the form $\pm(\boldsymbol{A} - zI_n)$ where $\boldsymbol{A}$ is the current matrix to be updated. It turns out, a slight change of perspective leads to iteration dependent matrix inversion. Namely, at iteration number $k$ the required $n \times n$ matrix inversions can be deduced from inverting $k \times k$ matrices (or better $m_k \times m_k$ matrices where $m_k$ is the number of unique past rank-one update). This improves the computational complexity for all iterations $k$ s.t. $m_k < n$. This allows sparsification techniques to be tractable even for large values of $n$, at least for the first iterations. The computational simplifications we propose are imperative for dual-set sparsification of sums of the form $\sum_{i=1}^m \boldsymbol{v}_i \boldsymbol{v}_i^\top \in \mathbb{R}^{n_1 \times n_1}$ and $\sum_{i=1}^m \boldsymbol{q}_i \boldsymbol{q}_i^\top \in \mathbb{R}^{n_2 \times n_2}$ where $n_2 >> n_1$. Indeed for a target number $N$ of rank-one update with $n_1 < N < n_2$, it is excessive to operate on matrices of size $n_2 \times n_2$ for the second sum. For single-set sparsification and in order to take full advantage of the identified workarounds, strategies based on restarting/aggregating every other "few" iterations are to be considered. We discuss a deterministic strategy that emulate lower barrier push as in Batson et al. (2009); Lee & Sun (2018) and which has shown promising results in the numerical experiment.

**Related works**    Improvement to spectral sparsification techniques are manifold and are concerned with many aspects. First, there are purely graph sparsifiers, concerned only with graph sparsification, though sophisticated edge sampling strategies Fung et al. (2011); Jambulapati & Sidford (2018), or in the presence of active constraints Koutis & Xu (2016); Kapralov et al. (2017); Arora & Upadhyay (2019). There are also improvements concerned with running time in the more general framework of sparsifying a sum of matrices $\boldsymbol{v}\boldsymbol{v}^\top$. For instance, through fast isotropic sparsification Zouzias, Anastasios (2012), random sampling and batch update rules Lee & Sun (2018), optimization grounded updates, mainly by means of semi-definite programming Allen-Zhu et al. (2015); Lee & Sun (2017); Cheng & Ge (2018), and optimized data structures for speeding up computations Song et al. (2022). The latter provides a comparaison on complexity and running time of these improvements.

Our contribution is not a parallel development to the aforementioned works, but rather a transversal one. The underlying ideas in this paper can be adapted to the frameworks such as Zouzias, Anastasios (2012); Lee & Sun (2018), and can benefit from optimized implementations like those in Song et al. (2022). For clarity of exposition, we focus on illustrating how these ideas apply specifically to the framework single-set and dual-set sparsification, Batson et al. (2009) and Boutsidis et al. (2011; 2013; 2014).

## 2 BSS FRAMEWORK

Let us recall the main theorem in Batson et al. (2009). We let $\boldsymbol{v}_1, \boldsymbol{v}_2, \ldots, \boldsymbol{v}_m$ be vectors in $\mathbb{R}^n$ and $\boldsymbol{M} = \sum_{i \leq m} \boldsymbol{v}_i \boldsymbol{v}_i^\top$. For every $\epsilon \in (0, 1)$, there exist scalars $s_i \geq 0$ with $|\{i : s_i \neq 0\}| \leq \lceil \mathrm{rank}(M)/\epsilon^2 \rceil$ s.t.

$$(1 - \epsilon)^2 \boldsymbol{M} \preceq \sum_{i \leq m} s_i \boldsymbol{v}_i \boldsymbol{v}_i^\top \preceq (1 + \epsilon)^2 \boldsymbol{M}$$

Up to consider vectors $\boldsymbol{w}_i = (\boldsymbol{M}^+)^{\frac{1}{2}} \boldsymbol{v}_i$ where $\boldsymbol{M}^+$ is the pseudo-inverse of $\boldsymbol{M}$ (or rather vectors $\boldsymbol{w}_i = \boldsymbol{L}^{-1} \boldsymbol{v}_i$ if $\boldsymbol{M}$ is nonsingular and $\boldsymbol{M} = \boldsymbol{L}\boldsymbol{L}^\top$ is a Cholesky decomposition of $\boldsymbol{M}$), it suffices to establishes the theorem for $\boldsymbol{M} = \boldsymbol{I}_n$ the identity matrix.

To prove the theorem, they build a sum $\boldsymbol{A} = \sum_i t_i \boldsymbol{v}_i \boldsymbol{v}_i^\top$ iteratively, adding one update $t_i \boldsymbol{v}_i \boldsymbol{v}_i^\top$ at a time that after $\lceil n/\epsilon^2 \rceil$ update satisfies $\lambda_{\max}(\boldsymbol{A})/\lambda_{\min}(\boldsymbol{A}) \leq (1 + \epsilon)^2/(1 - \epsilon)^2$. For $\boldsymbol{A}$ s.t. $\ell \boldsymbol{I}_n \prec \boldsymbol{A} \prec u \boldsymbol{I}_n$, we recall that lower/upper potentials are

$$\Phi_\ell(\boldsymbol{A}) \overset{\mathrm{def}}{=} \mathrm{Tr}((\boldsymbol{A} - \ell \boldsymbol{I}_n)^{-1}), \qquad \Phi^u(\boldsymbol{A}) \overset{\mathrm{def}}{=} \mathrm{Tr}((u \boldsymbol{I}_n - \boldsymbol{A})^{-1}) \tag{1}$$

Initially, $\boldsymbol{A} = \boldsymbol{0}$ and the barriers are at $\ell = \ell_0 < 0 < u_0 = u$. At each iteration, the matrix is updated by a rank-one matrix $t_i \boldsymbol{v}_i \boldsymbol{v}_i^\top$, that guarantees that while barriers $\ell$ and $u$ are incremented by $\delta_L$ and $\delta_U$, respectively, at each step, the lower and upper potentials do not increase. As a result, no eigenvalue ever jumps across a barrier.

More precisely, we let $N \geq 0$, $\epsilon_L, \epsilon_U, \delta_L, \delta_U > 0$ s.t. $1/\delta_U + \epsilon_U \leq 1/\delta_L - \epsilon_L$, and consider the following scheme

- Initialization: $\boldsymbol{A} = \boldsymbol{0}$, $u = n/\epsilon_U$ and $l = -n/\epsilon_L$, implying

$$\Phi_\ell(\boldsymbol{A}) = \epsilon_L, \quad \text{and} \quad \Phi^u(\boldsymbol{A}) = \epsilon_U.$$

- For $k = 1, \ldots, N$ do:
  - pick a vector $\boldsymbol{v} \in \{\boldsymbol{v}_i\}$ and $t \geq 0$ that insures

  $$\Phi_{\ell+\delta_L}(\boldsymbol{A} + t\boldsymbol{v}\boldsymbol{v}^\top) \leq \Phi_\ell(\boldsymbol{A}), \qquad \Phi^{u+\delta_U}(\boldsymbol{A} + t\boldsymbol{v}\boldsymbol{v}^\top) \leq \Phi^u(\boldsymbol{A}).$$

  - Update the matrix and increment the barrier $\ell$ and $u$,

  $$\boldsymbol{A} \leftarrow \boldsymbol{A} + t\boldsymbol{v}\boldsymbol{v}^\top, \qquad \ell \leftarrow \ell + \delta_L, \qquad u \leftarrow u + \delta_U.$$

The main difficulty in this sketched BSS algorithm resides in finding an adequate vector $\boldsymbol{v}$ and real number $t \geq 0$ s.t. the updated matrix yields a decrease in lower and upper potentials, with the new lower and upper barriers. This is however possible as thoroughly explained in Batson et al. (2009). We give a quick rundown of their arguments.

We assume that at the $k$-th iteration $\ell \boldsymbol{I}_n \prec \boldsymbol{A} \prec u \boldsymbol{I}_n$, $\Phi_\ell(\boldsymbol{A}) \leq \epsilon_L$ and $\Phi^u(\boldsymbol{A}) \leq \epsilon_U$. Then obviously $\boldsymbol{A} \prec (u + \delta_U)\boldsymbol{I}_n$ and since $\epsilon_L < 1/\delta_L$, one also has $(\ell + \delta_L)\boldsymbol{I}_n \prec \boldsymbol{A}$. For an arbitrary $\boldsymbol{v} \in \{\boldsymbol{v}_i\}$ and $t \geq 0$, applying Sherman-Morisson identity gives

$$\Phi_{\ell+\delta_L}(\boldsymbol{A} + t\boldsymbol{v}\boldsymbol{v}^\top) = \Phi_{\ell+\delta_L}(\boldsymbol{A}) - \frac{t\boldsymbol{v}^\top (\boldsymbol{A} - (\ell + \delta_L)\boldsymbol{I}_n)^{-2} \boldsymbol{v}}{1 + t\boldsymbol{v}^T (\boldsymbol{A} - (\ell + \delta_L)\boldsymbol{I}_n)^{-1} \boldsymbol{v}}.$$

$$\Phi^{u+\delta_U}(\boldsymbol{A} + t\boldsymbol{v}\boldsymbol{v}^\top) = \Phi^{u+\delta_U}(\boldsymbol{A}) + \frac{t\boldsymbol{v}^\top ((u + \delta_U)\boldsymbol{I}_n - \boldsymbol{A})^{-2} \boldsymbol{v}}{1 - t\boldsymbol{v}^T ((u + \delta_U)\boldsymbol{I}_n - \boldsymbol{A})^{-1} \boldsymbol{v}},$$

The second identity is justified if $D_U \neq 0$ where $D_U = (1 - t\boldsymbol{v}^T ((u + \delta_U)\boldsymbol{I}_n - \boldsymbol{A})^{-1} \boldsymbol{v})$. We note in passing that $\boldsymbol{A} + t\boldsymbol{v}\boldsymbol{v}^\top \prec (u + \delta_U)\boldsymbol{I}_n$ if and only if $D_U > 0$, which in turn constraints $t$ be in $[0, t^*[$ with $t^* = 1/\boldsymbol{v}^T ((u + \delta_U)\boldsymbol{I}_n - \boldsymbol{A})^{-1} \boldsymbol{v}$. We note that from lower/upper potentials definition,

$$\Phi_{\ell+\delta_L}(\boldsymbol{A}) > \Phi_\ell(\boldsymbol{A}), \qquad \Phi^{u+\delta_U}(\boldsymbol{A}) < \Phi^u(\boldsymbol{A}).$$

Also, note that $t \mapsto \Phi^{u+\delta_u}(\boldsymbol{A} + t\boldsymbol{v}\boldsymbol{v}^\top)$ strictly increases from $\Phi^{u+\delta_U}(\boldsymbol{A})$ to $+\infty$ for $t \in [0, t^*[$ and $t \mapsto \Phi_{\ell+\delta_L}(\boldsymbol{A} + t\boldsymbol{v}\boldsymbol{v}^\top)$ is strictly decreasing. As $t$ is increased, one is faced with the opposed

objectives of dropping the lower potential below $\Phi_\ell(\boldsymbol{A})$ while also keeping the upper potential below $\Phi^u(\boldsymbol{A})$. If we cast these objectives as equations on $t > 0$, we obtain $1/t \leq L_{\boldsymbol{A}}(\boldsymbol{v})$ and $U_{\boldsymbol{A}}(\boldsymbol{v}) \leq 1/t$ where

$$
\begin{aligned}
U_{\boldsymbol{A}}(\boldsymbol{v}) &\stackrel{\text{def}}{=} \frac{\boldsymbol{v}^T\left((u+\delta_U)\,\boldsymbol{I}_n - \boldsymbol{A}\right)^{-2}\boldsymbol{v}}{\Phi^u(\boldsymbol{A}) - \Phi^{u+\delta_U}(\boldsymbol{A})} + \boldsymbol{v}^T\left((u+\delta_U)\,\boldsymbol{I}_n - \boldsymbol{A}\right)^{-1}\boldsymbol{v}, \\
L_{\boldsymbol{A}}(\boldsymbol{v}) &\stackrel{\text{def}}{=} \frac{\boldsymbol{v}^T\left(\boldsymbol{A} - (\ell+\delta_L)\,\boldsymbol{I}_n\right)^{-2}\boldsymbol{v}}{\Phi_{\ell+\delta_L}(\boldsymbol{A}) - \Phi_\ell(\boldsymbol{A})} - \boldsymbol{v}^T\left(\boldsymbol{A} - (\ell+\delta_L)\,\boldsymbol{I}_n\right)^{-1}\boldsymbol{v}.
\end{aligned}
\tag{2}
$$

These quantities are well defined for all $\boldsymbol{v} \in \{\boldsymbol{v}_i\}$ with $U_{\boldsymbol{A}}(\boldsymbol{v}) > 0$ for all $\boldsymbol{v}$. We note also that the condition $U_{\boldsymbol{A}}(\boldsymbol{v}) \leq 1/t$ implies necessarily $t \in [0, t^*[$. The authors in Batson et al. (2009) prescribe naturally picking any $\boldsymbol{v}$ and $t$ s.t. $U_{\boldsymbol{A}}(\boldsymbol{v}) \leq 1/t \leq L_{\boldsymbol{A}}(\boldsymbol{v})$. They indeed demonstrated by an averaging argument that the inequality $U_{\boldsymbol{A}}(\boldsymbol{v}) \leq L_{\boldsymbol{A}}(\boldsymbol{v})$ must holds for at least one $\boldsymbol{v}$. For this to hold, the conditions $1/\delta_U + \epsilon_U \leq 1/\delta_L - \epsilon_L$ is sufficient as it implies $\sum_{\boldsymbol{v}} U_{\boldsymbol{A}}(\boldsymbol{v}) \leq \sum_{\boldsymbol{v}} L_{\boldsymbol{A}}(\boldsymbol{v})$.

Given $\epsilon \in ]0, 1[$, we let $\kappa = \frac{1+\epsilon}{1-\epsilon}$ and consider parameters

$$
\delta_L = 1, \qquad \epsilon_L = \epsilon, \qquad \delta_U = \kappa, \qquad \epsilon_U = \epsilon/\kappa.
\tag{3}
$$

One has $1/\delta_U + \epsilon_U = (1+\epsilon)/\kappa = 1 - \epsilon = 1/\delta_L - \epsilon_L$. Running the BSS algorithm for $N$ iteration yields $\boldsymbol{A} = \sum_i t_i \boldsymbol{v}_i \boldsymbol{v}_i^\top$ with $|\{i : t_i \neq 0\}| \leq N$ and

$$
(-n/\epsilon + N)\boldsymbol{I}_n \prec \boldsymbol{A} \prec \kappa(n/\epsilon + N)\boldsymbol{I}_n.
\tag{4}
$$

For $N \geq n/\epsilon$, the matrix $\boldsymbol{A}$ is guaranteed definite positive, and for $N = (1+\gamma)n/\epsilon$ it satisfies in addition $\lambda_{\max}(\boldsymbol{A})/\lambda_{\min}(\boldsymbol{A}) \leq \kappa\frac{2+\gamma}{\gamma}$. For $N = \lceil n/\epsilon^2 \rceil$ (hence $\gamma \approx (\epsilon^{-1} - 1)$). Normalizing $\boldsymbol{A}$ by the lower barrier $(-n/\epsilon + N) \approx n(1-\epsilon)/\epsilon^2$ and using the fact that $(n/\epsilon + x)/(-n/\epsilon + x)$ is strictly decreasing for $x > 0$, yields $\boldsymbol{I}_n \prec \boldsymbol{A} \prec \kappa^2 \boldsymbol{I}_n$.

A sketch of the constructive proof is presented in Algorithm 1.

---

**Algorithm 1** Single set sparsification algorithm

---

**Require:** $\{\boldsymbol{v}_i\}_{i=1}^m$ s.t. $\sum_{i=1}^m \boldsymbol{v}_i \boldsymbol{v}_i^\top = \boldsymbol{I}_n$, $N > n$,
**Ensure:** $\boldsymbol{A} = \sum_i t_i \boldsymbol{v}_i \boldsymbol{v}_i^\top$ s.t. $|\{i : t_i \neq 0\}| \leq N$, and $(1 - \sqrt{n/N})^2 \boldsymbol{I}_n \prec \boldsymbol{A} \prec (1 + \sqrt{n/N})^2 \boldsymbol{I}_n$
 1: Let $\epsilon = \sqrt{n/N}$, $\kappa = (1+\epsilon)/(1-\epsilon)$, and

$$
\delta_L = 1, \qquad \epsilon_L = \epsilon, \qquad \delta_U = \kappa, \qquad \epsilon_U = \epsilon/\kappa.
$$

 2: Initialize $\boldsymbol{A} = \boldsymbol{0}_{n \times n}$, $l = -n/\epsilon_L$, $u = n/\epsilon_U$, $\Phi_\ell = \epsilon_L$, $\Phi_u = \epsilon_U$
 3: **for** $k = 1, \ldots, N$ **do**
 4:     Select $\boldsymbol{v} \in \{\boldsymbol{v}_i\}$ and weights $t > 0$ satisfying

$$
U\left(\boldsymbol{A}, u, \delta_U, \boldsymbol{v}\right) \leq 1/t \leq L\left(\boldsymbol{A}, \ell, \delta_L, \boldsymbol{v}\right).
$$

 5:     update $\boldsymbol{A} \leftarrow \boldsymbol{A} + t\boldsymbol{v}\boldsymbol{v}^\top$, $l \leftarrow l + \delta_L$, $u \leftarrow u + \delta_U$,
 6: **end for**
 7: multiply selected weights $t$ and $\boldsymbol{A}$ by $(1 - \sqrt{n/N})N^{-1}$

---

## 3 DUAL-SET SPARSIFICATION FRAMEWORK

The exact same constructive proof can be considered for a dual-set sparsification setting, see e.g. Boutsidis et al. (2014). Namely, let $\boldsymbol{v}_1, \boldsymbol{v}_2, \ldots, \boldsymbol{v}_m$ be vectors in $\mathbb{R}^{n_1}$ and $\boldsymbol{q}_1, \boldsymbol{q}_2, \ldots, \boldsymbol{q}_m$ vectors in $\mathbb{R}^{n_2}$ s.t. $\sum_{i \leq m} \boldsymbol{v}_i \boldsymbol{v}_i^\top = \boldsymbol{I}_{n_1}$ and $\sum_{i \leq m} \boldsymbol{q}_i \boldsymbol{q}_i^\top = \boldsymbol{I}_{n_2}$. For every $N > n_1$, there exist scalars $s_i \geq 0$ with $|\{i : s_i \neq 0\}| \leq N$ s.t.

$$
(1 - \sqrt{n_1/N})^2 \boldsymbol{I}_{n_1} \preceq \sum_{i \leq m} s_i \boldsymbol{v}_i \boldsymbol{v}_i^\top, \qquad \sum_{i \leq m} s_i \boldsymbol{q}_i \boldsymbol{q}_i^\top \preceq (1 + \sqrt{n_1/N})^2 \boldsymbol{I}_{n_2}
$$

A sketch of the constructive proof is presented in Algorithm 2.

---

**Algorithm 2** Dual set sparsification algorithm

---

**Require:** $\{v_i\}_{i=1}^m$ s.t. $\sum_{i=1}^m v_i v_i^\top = \mathbf{I}_{n_1}$, $\{\mathbf{q}_i\}_{i=1}^m$ s.t. $\sum_{i=1}^m \mathbf{q}_i \mathbf{q}_i^\top = \mathbf{I}_{n_2}$, $N > n_1$,
**Ensure:** $\boldsymbol{A} = \sum_i t_i v_i v_i^\top$, $\boldsymbol{H} = \sum_i t_i \boldsymbol{q}_i \boldsymbol{q}_i^\top$ s.t. $|\{i : t_i \neq 0\}| \leq N$, and

$$(1 - \sqrt{n_1/N})^2 \boldsymbol{I}_n \preceq \boldsymbol{A}, \qquad \boldsymbol{H} \preceq (1 + \sqrt{n_2/N})^2 \boldsymbol{I}_n$$

1: Let $\epsilon_1 = \sqrt{n_1/N}$, $\epsilon_2 = \sqrt{n_2/N}$, $\kappa = (1 + \epsilon_2)/(1 - \epsilon_1)$, and

$$\delta_L = 1, \qquad \epsilon_L = \epsilon_1, \qquad \delta_U = \kappa, \qquad \epsilon_U = \epsilon_2/\kappa.$$

2: Initialize $\boldsymbol{A} = \mathbf{0}_{n_1 \times n_1}$, $\boldsymbol{H} = \mathbf{0}_{n_2 \times n_2}$, $l = -n_1/\epsilon_L$, $u = n_2/\epsilon_U$, $\Phi_\ell = \epsilon_L$, $\Phi_u = \epsilon_U$
3: **for** $k = 1, \ldots, N$ **do**
4:     Select couple $(\boldsymbol{v}, \mathbf{q}) \in \{(\boldsymbol{v}_i, \mathbf{q}_i)\}$ and weights $t > 0$ satisfying

$$U(\boldsymbol{H}, u, \delta_U, \boldsymbol{q}) \leq 1/t \leq L(\boldsymbol{A}, \ell, \delta_L, \boldsymbol{v}).$$

5:     update $\boldsymbol{A} \leftarrow \boldsymbol{A} + t\boldsymbol{v}\boldsymbol{v}^\top$, $\boldsymbol{H} \leftarrow \boldsymbol{H} + t\boldsymbol{v}\boldsymbol{v}^\top$, $l \leftarrow l + \delta_L$, $u \leftarrow u + \delta_U$,
6: **end for**
7: multiply selected weights $t$, $\boldsymbol{A}$ and $\boldsymbol{H}$ by $(1 - \sqrt{n_1/N})N^{-1}$

---

## 4    COMPUTATIONAL IMPROVEMENTS

In Algorithms 1 and 2. Every iteration is dominated by the computation of $\mathrm{Tr}(\boldsymbol{M}^{-1})$ and evaluations $\boldsymbol{v}\boldsymbol{M}^{-1}\boldsymbol{v}$ and $\boldsymbol{v}\boldsymbol{M}^{-2}\boldsymbol{v}$ for $\boldsymbol{M}$ the two matrices $(\boldsymbol{A} - (l + \delta_L)\boldsymbol{I}_n)$ and $((u + \delta_U)\boldsymbol{I}_n - \boldsymbol{H})$. [1]. In plain algorithm description, it is usually assumed that these inverses are computed at the beginning of every iteration and required computations are carried out in the most natural and direct manner. Here, we present the simplification (workarounds) for such computations. They improve greatly the speed of the algorithms on the first $n$ or more iterations and are imperative dual sparsification in the case $n_1 < N < n_2$.

**Few linear algebra lemmas**    The Woodbury matrix identity is

$$(\boldsymbol{A} + \boldsymbol{X}\boldsymbol{C}\boldsymbol{Y})^{-1} = \boldsymbol{A}^{-1} - \boldsymbol{A}^{-1}\boldsymbol{X}(\boldsymbol{C}^{-1} + \boldsymbol{Y}\boldsymbol{A}^{-1}\boldsymbol{X})^{-1}\boldsymbol{Y}\boldsymbol{A}^{-1}$$

where $\boldsymbol{A}, \boldsymbol{C}, \boldsymbol{X}$ and $\boldsymbol{Y}$ are conformable matrices: $\boldsymbol{A}$ is $n \times n$, $\boldsymbol{C}$ is $k \times k$, $\boldsymbol{X}$ is $n \times k$, $\boldsymbol{Y}$ is $k \times n$ and the inverses are assumed to be well defined. The Weinstein–Aronszajn identity states

$$\det(\boldsymbol{I}_n + \boldsymbol{X}\boldsymbol{Y}) = \det(\boldsymbol{I}_k + \boldsymbol{Y}\boldsymbol{X}).$$

By an immediate application to $\boldsymbol{A} = -z\boldsymbol{I}_n$, $\boldsymbol{C} = \boldsymbol{I}_k$ and $\boldsymbol{Y} = \boldsymbol{X}^\top$, we have the following lemma.

**Lemma 1.** *Let* $\boldsymbol{X} \in \mathbb{R}^{n \times k}$, *and* $z \in \mathbb{R} - \{0\}$. *Then* $(\boldsymbol{X}\boldsymbol{X}^\top - z\boldsymbol{I}_n)$ *is nonsingular if and only if* $(\boldsymbol{X}^\top\boldsymbol{X} - z\mathbf{I}_k)$ *is nonsingular. Moreover, there holds*

$$(\boldsymbol{X}\boldsymbol{X}^\top - z\boldsymbol{I}_n)^{-1} = \frac{\boldsymbol{X}(\boldsymbol{X}^\top\boldsymbol{X} - z\mathbf{I}_k)^{-1}\boldsymbol{X}^\top - \boldsymbol{I}_n}{z} \tag{5}$$

This lemma can also be derived via SVD decomposition of $\boldsymbol{X}$. Taking the square of the identity and performing some simplifications, we derive the following lemma.

**Lemma 2.** *Let* $\boldsymbol{X} \in \mathbb{R}^{n \times k}$, *and* $z \in \mathbb{R} - \{0\}$ *s.t.* $\boldsymbol{X}\boldsymbol{X}^\top - z I_k$ *is nonsingular. There holds*

$$(\boldsymbol{X}\boldsymbol{X}^\top - z\boldsymbol{I}_n)^{-2} + \frac{(\boldsymbol{X}\boldsymbol{X}^\top - z\boldsymbol{I}_n)^{-1}}{z} = \frac{\boldsymbol{X}(\boldsymbol{X}^\top\boldsymbol{X} - z\mathbf{I}_k)^{-2}\boldsymbol{X}^\top}{z} \tag{6}$$

*Proof.* We have that

$$\left(\boldsymbol{X}(\boldsymbol{X}^\top\boldsymbol{X} - z\mathbf{I}_k)^{-1}\boldsymbol{X}^\top\right)^2 = \boldsymbol{X}(\boldsymbol{X}^\top\boldsymbol{X} - z\mathbf{I}_k)^{-1}\boldsymbol{X}^\top\boldsymbol{X}(\boldsymbol{X}^\top\boldsymbol{X} - z\mathbf{I}_k)^{-1}\boldsymbol{X}^\top$$

$$= z\boldsymbol{X}(\boldsymbol{X}^\top\boldsymbol{X} - z\mathbf{I}_k)^{-2}\boldsymbol{X}^\top + \boldsymbol{X}(\boldsymbol{X}^\top\boldsymbol{X} - z\mathbf{I}_k)^{-1}\boldsymbol{X}^\top,$$

where we have simply used $\boldsymbol{X}^\top\boldsymbol{X} = z\boldsymbol{I}_n + (\boldsymbol{X}^\top\boldsymbol{X} - z\boldsymbol{I}_n)$. Taking the square of equation 5 and rearranging the right hand side using the above identity, we derive the claimed result. $\qquad\square$

---

[1] $\boldsymbol{H} = \boldsymbol{A}$ for single-set sparsification

An immediate implication of Lemmas 1 and 2 is the following.

**Corollary 1.** *Let $\boldsymbol{X} \in \mathbb{R}^{n \times k}$, $\boldsymbol{A} = \boldsymbol{X}\boldsymbol{X}^\top$, $\boldsymbol{B} = \boldsymbol{X}^\top\boldsymbol{X}$ and $z \neq 0$ s.t. $\boldsymbol{A} - z\mathbf{I}_k$ is non nonsingular. For $\boldsymbol{v} \in \mathbb{R}^n$, there holds*

$$\boldsymbol{v}^\top (\boldsymbol{A} - z\boldsymbol{I}_n)^{-1} \boldsymbol{v} = \frac{\boldsymbol{w}^\top (\boldsymbol{B} - z\mathbf{I}_k)^{-1} \boldsymbol{w} - \|\boldsymbol{v}\|^2}{z}, \tag{7}$$

*where $\boldsymbol{w} = \boldsymbol{X}^\top\boldsymbol{v}$. We let $q_-(\boldsymbol{v})$ be the value, then*

$$\boldsymbol{v}^\top (\boldsymbol{A} - z\boldsymbol{I}_n)^{-2} \boldsymbol{v} = \frac{\boldsymbol{w}^\top (\boldsymbol{B} - z\mathbf{I}_k)^{-2} \boldsymbol{w} - q_-(\boldsymbol{v})}{z} \tag{8}$$

**Fast potentials and quadratic forms computation** Sparsification algorithms as discussed consist in picking a new pair of vector/weight $(\boldsymbol{v}_{i_k}, t_k)$ at the $k$-th iteration and update $\boldsymbol{A} \leftarrow \boldsymbol{A} + t_k \boldsymbol{v}_{i_k} \boldsymbol{v}_{i_k}^\top$ or $(\boldsymbol{v}_{i_k}, \boldsymbol{q}_{i_k}, t_k)$ at the $k$-th iteration and update $\boldsymbol{A} \leftarrow \boldsymbol{A} + t_k \boldsymbol{v}_{i_k} \boldsymbol{v}_{i_k}^\top$ and $\boldsymbol{H} \leftarrow \boldsymbol{H} + t_k \boldsymbol{q}_{i_k} \boldsymbol{q}_{i_k}^\top$ for dual set sparsification. We introduce notation $\boldsymbol{X} = [\sqrt{t_1}\boldsymbol{v}_{i_1} | \ldots | \sqrt{t_k}\boldsymbol{v}_{i_k}] \in \mathbb{R}^{n \times k}$. Then at iteration $k$, one has

$$\boldsymbol{A} = \sum_{j=1}^{k} t_j \boldsymbol{v}_{i_j} \boldsymbol{v}_{i_j}^\top = \boldsymbol{X}\boldsymbol{X}^\top$$

The matrix $\boldsymbol{A}$ has the same eigenvalues as the matrix $\boldsymbol{B} = \boldsymbol{X}^\top\boldsymbol{X} (\in \mathbb{R}^{k \times k})$. More precisely, if $\lambda_1 \geq \cdots \geq \lambda_n \geq 0$ are the eigenvalues of $\boldsymbol{A}$, then $\lambda_1 \geq \cdots \geq \lambda_k \geq 0$ are the eigenvalues $\boldsymbol{B}$ (with $\lambda_{n+1} = \cdots = \lambda_k$ if $k \geq n$ and $\lambda_{k+1} = \cdots = \lambda_n = 0$ if $k \leq n$). In particular, if $l, u \in \mathbb{R}$ are such that $lI_n \prec \boldsymbol{A} \prec uI_n$, then $lI_k \prec \boldsymbol{B} \prec uI_k$, and

$$\Phi_\ell(\boldsymbol{A}) = \Phi_\ell(\boldsymbol{B}) - \frac{n-k}{l}, \qquad \Phi^u(\boldsymbol{A}) = \Phi^u(\boldsymbol{B}) + \frac{n-k}{u}. \tag{9}$$

Quadratic forms associated with matrices of the form $(\boldsymbol{A} - z\mathbf{I})^{-1}$, $(\boldsymbol{A} - z\mathbf{I})^{-2}$, $(z\mathbf{I} - \boldsymbol{A})^{-1}$, and $(z\mathbf{I} - \boldsymbol{A})^{-2}$ are related to those same quadratic forms but associated with $\boldsymbol{B}$, see Corollary 1. Let us present this for our settings of $k$-th iteration of single-set or dual-set sparsification algorithm and assume that

$$(l + \delta_L)\,\boldsymbol{I}_n \prec \boldsymbol{A} \prec (u + \delta_U)\,\boldsymbol{I}_n.$$

Then, $(l + \delta_L)\,\mathbf{I}_k \prec \boldsymbol{B} \prec (u + \delta_U)\,\mathbf{I}_k$. Moreover, given $\boldsymbol{v} \in \mathbb{R}^n$ and introducing $\boldsymbol{w} = \boldsymbol{X}^\top\boldsymbol{v} (\in \mathbb{R}^k)$,

$$Q_{L,1}(\boldsymbol{v}) = \boldsymbol{v}^\top (\boldsymbol{A} - (l + \delta_L)\,\mathbf{I})^{-1} \boldsymbol{v} = \frac{\boldsymbol{w}^\top (\boldsymbol{B} - (l + \delta_L)\,\mathbf{I}_k)^{-1} \boldsymbol{w} - \|\boldsymbol{v}\|^2}{l + \delta_L}, \tag{10}$$

$$Q_{L,2}(\boldsymbol{v}) = \boldsymbol{v}^\top (\boldsymbol{A} - (l + \delta_L)\,\mathbf{I})^{-2} \boldsymbol{v} = \frac{\boldsymbol{w}^\top (\boldsymbol{B} - (l + \delta_L)\,\mathbf{I}_k)^{-2} \boldsymbol{w} - Q_{L,1}(\boldsymbol{v})}{l + \delta_L}. \tag{11}$$

and

$$Q_{U,1}(\boldsymbol{v}) = \boldsymbol{v}^\top ((u + \delta_U)\,\mathbf{I} - \boldsymbol{A})^{-1} \boldsymbol{v} = \frac{\boldsymbol{w}^\top ((u + \delta_U)\,\mathbf{I}_k - \boldsymbol{B})^{-1} \boldsymbol{w} + \|\boldsymbol{v}\|^2}{u + \delta_U}, \tag{12}$$

$$Q_{U,2}(\boldsymbol{v}) = \boldsymbol{v}^\top ((u + \delta_U)\,\mathbf{I} - \boldsymbol{A})^{-2} \boldsymbol{v} = \frac{\boldsymbol{w}^\top ((u + \delta_U)\,\mathbf{I}_k - \boldsymbol{B})^{-1} \boldsymbol{w} + Q_{L,1}(\boldsymbol{v})}{u + \delta_U}. \tag{13}$$

In particular, the knowledge of matrices $\boldsymbol{X}$ and $\boldsymbol{B}$ is enough to compute quantities $L(\boldsymbol{A}, \ell, \delta_L, \boldsymbol{v})$ and $U(\boldsymbol{A}, u, \delta_U, \boldsymbol{v})$.

We assume we have matrices $\boldsymbol{X} \in \mathbb{R}^{n \times k}$, $\boldsymbol{A} = \boldsymbol{X}\boldsymbol{X}^\top$, and $\boldsymbol{B} = \boldsymbol{X}^\top\boldsymbol{X}$ as described above. We summarize below the time complexities for computing matrix inverse $(\boldsymbol{A} - z\mathbf{I}_k)^{-1}$ vs $(\boldsymbol{B} - z\mathbf{I}_k)^{-1}$, for computing their traces, for matrix-vector multiplication $(\boldsymbol{A} - z\mathbf{I}_k)^{-1} \boldsymbol{v}$ vs $(\boldsymbol{B} - z\mathbf{I}_k)^{-1} \boldsymbol{w}$ with $\boldsymbol{w} = \boldsymbol{X}^\top\boldsymbol{v}$, for computing scalar products (and squared euclidean norm) and finally for computing quantities such as $L(\boldsymbol{A}, \ell, \delta_L, \boldsymbol{v})$ and $U(\boldsymbol{A}, u, \delta_U, \boldsymbol{v})$ relying on $\boldsymbol{A}$ vs relying on $\boldsymbol{B}$.

We note that in the actual sparsification algorithms, the complexity for computing vectors $\boldsymbol{w} = \boldsymbol{X}^\top\boldsymbol{v}$ for $\boldsymbol{v} \in \{\boldsymbol{v}_i\}_{i=1}^m$ can be reduced to $\mathcal{O}(1)$ by storing computation from previous iterations. In matrix-vector multiplication below, we take this into account, and have $\mathcal{O}(k^2)$ instead of $\mathcal{O}(n) + \mathcal{O}(k^2)$.

In light of the comparaison table, we have the following theorem on Algorithm 2 improved complexity.

| Operations \ Algorithm | relying on $\boldsymbol{A}$ | relying on $\boldsymbol{B}$ |
|---|---|---|
| Matrix inversion | $\mathcal{O}(n^3)$ | $\mathcal{O}(k^3)$ |
| Trace computation | $\mathcal{O}(n)$ | $\mathcal{O}(k)$ |
| Matrix-vector multiplication | $\mathcal{O}(n^2)$ | $+\mathcal{O}(k^2)$ |
| scalar product/norm squared | $\mathcal{O}(n)$ | $\mathcal{O}(k)$ |
| $L_{\boldsymbol{A}}(\boldsymbol{v})$ and $U_{\boldsymbol{A}}(\boldsymbol{v})$ for all $\boldsymbol{v} \in \{\boldsymbol{v}_i\}_{i=1}^m$ | $\mathcal{O}(n^3 + mn^2)$ | $\mathcal{O}(k^3 + mk^2)$ |

**Theorem 1.** *Let* $\boldsymbol{v}_1, \boldsymbol{v}_2, \ldots, \boldsymbol{v}_m$ *be vectors in* $\mathbb{R}^{n_1}$ *and* $\boldsymbol{q}_1, \boldsymbol{q}_2, \ldots, \boldsymbol{q}_m$ *vectors in* $\mathbb{R}^{n_2}$ *s.t.* $\sum_{i \leq m} \boldsymbol{v}_i \boldsymbol{v}_i^\top = \boldsymbol{I}_{n_1}$ *and* $\sum_{i \leq m} \boldsymbol{q}_i \boldsymbol{q}_i^\top = \boldsymbol{I}_{n_2}$ *and assume that* $n_1 \leq n_2$. *For every* $m > N > n_1$, *there exist scalars* $s_i \geq 0$ *with* $|\{i : s_i \neq 0\}| \leq N$ *s.t.*

$$(1 - \sqrt{n_1/N})^2 \boldsymbol{I}_{n_1} \preceq \sum_{i \leq m} s_i \boldsymbol{v}_i \boldsymbol{v}_i^\top, \qquad \sum_{i \leq m} s_i \boldsymbol{q}_i \boldsymbol{q}_i^\top \preceq (1 + \sqrt{n_1/N})^2 \boldsymbol{I}_{n_2}$$

*The sparsification algorithm runs in* $\mathcal{O}(Nmn_1^2) + \mathcal{O}(Nm \min(n_2^2, N^2))$

The computational complexity $\mathcal{O}(Nm \min(n_2^2, N^2))$ follows as the minimum of complexities $\mathcal{O}(Nmn_2^2)$ and $\mathcal{O}(mN^3)$ which corresponds to Algorithm 2 improved complexity. in the case $n_2 > N$ and $N \leq n_2$.

## 5 CONCLUSION

We have presented a transversal contribution that extends core ideas from spectral sparsification. Our framework offers both theoretical insight and practical flexibility, with potential applications across graph theory, linear algebra, and machine learning. Future endeavor will primarily focus on establishing theoretical guarantees for the restarted/aggregated framework.

**Reproducibility Statement.** Reproducibility is supported by: clear problem setup, notation, and assumptions in Section 2 and 3 and complete or sketched proofs. Implementation details and experimental settings for computing sparse sums are revisited and detailed in the appendix.

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

## A   APPENDIX: ALGORITHMS AND ADDITIONAL DETAILS

Here we revisit in more details the single-set and dual-set sparsification frameworks discussed in the paper. An implementation of the BSS algorithm is provided in Algorithm 1. Every iteration is dominated by the computation of $\mathrm{Tr}(\boldsymbol{M}^{-1})$ and evaluations $\boldsymbol{v}\boldsymbol{M}^{-1}\boldsymbol{v}$ and $\boldsymbol{v}\boldsymbol{M}^{-2}\boldsymbol{v}$ for $\boldsymbol{M}$ the two matrices $(\boldsymbol{A} - (l + \delta_L)\boldsymbol{I}_n)$ and $((u + \delta_U)\boldsymbol{I}_n - \boldsymbol{A})$. In the plain algorithm description, we simply assume we compute these inverses $\boldsymbol{Z}_l$ and $\boldsymbol{Z}_u$ at the beginning of every iteration and carry out required computations in the most natural and direct manner. One other more stable and convenient way to perform this is by means of Cholesky decomposition. Given $\boldsymbol{M}$ an $n \times n$ symmetric definite positive and $\boldsymbol{M} = \boldsymbol{L}\boldsymbol{L}^{\top}$ its cholesky decomposition, there holds $\mathrm{Tr}(\boldsymbol{M}^{-1}) = \|\boldsymbol{L}^{-1}\|_F^2$ and

$$\boldsymbol{v}^{\top}\boldsymbol{M}^{-1}\boldsymbol{v} = \|\boldsymbol{L}^{-1}\boldsymbol{v}\|^2, \qquad \boldsymbol{v}^{\top}\boldsymbol{M}^{-2}\boldsymbol{v} = \|(\boldsymbol{L}^{-1})^{\top}\boldsymbol{L}^{-1}\boldsymbol{v}\|^2. \tag{14}$$

The quantities $\Phi_{\ell + \delta_L}$, $\Phi^{u + \delta_U}$ and the evaluations $q_{l,1}, q_{l,2}, q_{u,1}, q_{u,2}$ in Algorithm equation 3 can thus be computed by relying on Cholesky decomposition with $\boldsymbol{M}$ equal to $(\boldsymbol{A} - (\ell + \delta_L)\boldsymbol{I}_n)$ or $((u + \delta_U)\boldsymbol{I}_n - \boldsymbol{A})$. One needs to perform two Cholesky decomposition and two matrix inversion of the lower matrices at the beginning of every iteration, the other operations are straightforward.

For the improved and faster computation of potentials and quadratic forms associated with $\boldsymbol{B}$, Cholesky decomposition can be invoked exactly as explained above.

In both Algorithm 3 and Algorithm 5, we can keep track on "unlocked" indices $i_k$, i.e. indices for which $(\boldsymbol{v}_{i_k}, t_k)$ was selected prior to the $k$-th iteration. We note that $m_k := |\{i_j : 1 \le j \le k| \le k$ since vectors can be reselected. In Algorithm 5 if $\boldsymbol{v}$ was selected in a previous iteration, we can simply update associated column in $\boldsymbol{X}$ replacing $\sqrt{t_{old}}$ with $\sqrt{t_{old} + t_{new}}$ and also reflect this on $\boldsymbol{B}$. In case this detail is implemented, Algorithm 5 is faster than Algorithm 3 on all iteration $k$ s.t. $m_k \le n$.

In the improved algorithm 5, $\boldsymbol{X}$ is only needed to compute the output $\boldsymbol{A} = \boldsymbol{X}\boldsymbol{X}^{\top}$ if the latter is not iteratively updated. We can dismiss it in the implementation and simply iteratively update $\boldsymbol{A} \leftarrow \boldsymbol{A} + t\boldsymbol{v}\boldsymbol{v}^{\top}$ initialized at $\boldsymbol{A} = \boldsymbol{0}_{n \times n}$. For applications where the knowledge of the final weights $\{t_1, \ldots, t_m\}$ is required, it is straightforward to implement the updating rule.

The improved implementation of dual-set sparsification is not fully detailed. However, the underlying ideas for simplification/speed up/caching are as demonstrated in Algorithm 5

## A.1 PLAIN SINGLE SET SPARSIFICATION ALGORITHM

For implementing plain single-set sparsification, we replace the generic instruction at line 6 and computations at lines 7,8 in Algorithm 3 with the more detailed subroutine 4.

---

**Algorithm 3** Plain single set sparsification algorithm

---

**Require:** $\{\boldsymbol{v}_i\}_{i=1}^m$ s.t. $\sum_{i=1}^n \boldsymbol{v}_i \boldsymbol{v}_i^\top = \boldsymbol{I}_n$, $N \geq n$,

**Ensure:** $\boldsymbol{A} = \sum_i t_i \boldsymbol{v}_i \boldsymbol{v}_i^\top$ s.t. $|\{i : t_i \neq 0\}| \leq r$, and $(1 - \sqrt{n/N})^2 \boldsymbol{I}_n \prec \boldsymbol{A} \prec (1 + \sqrt{n/N})^2 \boldsymbol{I}_n$

1: Let $\epsilon = \sqrt{n/N}$, $\kappa = (1 + \epsilon)/(1 - \epsilon)$, and

$$\delta_L = 1, \qquad \epsilon_L = \epsilon, \qquad \delta_U = \kappa, \qquad \epsilon_U = \epsilon/\kappa.$$

2: Initialize $\boldsymbol{A} = \boldsymbol{0}_{n \times n}$,           ▷ matrices initialization

3: Initialize $\ell = -n/\epsilon_L$,   $u = n/\epsilon_U$,           ▷ barriers initialization

4: Initialize $\Phi_\ell = \epsilon_L$,   $\Phi_u = \epsilon_U$,           ▷ potentials initialization

5: **for** $k = 1, \ldots, N$ **do**

6:      select vector $\boldsymbol{v} \in \{\boldsymbol{v}_i\}$ and number $t > 0$ satisfying

$$U(\boldsymbol{v}, \delta_U, \boldsymbol{A}, u) \leq \frac{1}{t} \leq L(\boldsymbol{v}, \delta_L, \boldsymbol{A}, \ell).$$

7:      compute $q_{\ell,1} = \boldsymbol{v}^\top (\boldsymbol{A} - (\ell + \delta_L)\boldsymbol{I}_n)^{-1} \boldsymbol{v}$,    $q_{\ell,2} = \boldsymbol{v}^\top (\boldsymbol{A} - (\ell + \delta_L)\boldsymbol{I}_n)^{-2} \boldsymbol{v}$

8:      compute $q_{u,1} = \boldsymbol{v}^\top ((u + \delta_U)\boldsymbol{I}_n - \boldsymbol{A})^{-1} \boldsymbol{v}$,    $q_{u,2} = \boldsymbol{v}^\top ((u + \delta_U)\boldsymbol{I}_n - \boldsymbol{A})^{-2} \boldsymbol{v}$

9:      update

$$\Phi_\ell \leftarrow \Phi_{\ell+\delta_L} + \frac{q_{l,2}}{1/t - q_{l,1}}, \qquad \Phi_u \leftarrow \Phi_{u+\delta_u} - \frac{q_{u,2}}{1/t + q_{u,1}}$$

10:      update $\boldsymbol{A} \leftarrow \boldsymbol{A} + t\boldsymbol{v}\boldsymbol{v}^\top$,   $\ell \leftarrow \ell + \delta_L$,   $u \leftarrow u + \delta_U$,

11: **end for**

12: multiply selected weights $t$ and $\boldsymbol{A}$ by $(1 - \sqrt{n/N})N^{-1}$

---

**Algorithm 4** selection of vector/weight $(\boldsymbol{v}, t)$

---

1: compute $\boldsymbol{Z}_\ell = (\boldsymbol{A} - (\ell + \delta_L)\boldsymbol{I}_n)^{-1}$, $\Phi_{\ell+\delta_L} = \mathrm{Tr}(\boldsymbol{Z}_\ell)$, and $\Delta_\ell = \Phi_\ell - \Phi_{\ell+\delta_L}$

2: compute $\boldsymbol{Z}_u = ((u + \delta_U)\boldsymbol{I}_n - \boldsymbol{A})^{-1}$, $\Phi_{u+\delta_U} = \mathrm{Tr}(\boldsymbol{Z}_u)$, and $\Delta_u = \Phi_{u+\delta_U} - \Phi_u$

3: consider variables $q_{l,1}, q_{l,2}, q_{u,1}, q_{u,2}, L, U$

4: **for** $i = 1$ to $m$ **do**

5:      Let $\boldsymbol{v} = \boldsymbol{v}_i$ and compute $\boldsymbol{x}_\ell = \boldsymbol{Z}_\ell \boldsymbol{v}$, and $\boldsymbol{x}_u = \boldsymbol{Z}_u \boldsymbol{v}$

6:      compute

$$q_{l,1} \leftarrow \langle \boldsymbol{v}, \boldsymbol{x}_l \rangle, \qquad q_{l,2} \leftarrow \|\boldsymbol{x}_l\|^2, \qquad L \leftarrow q_{l,2}/\Delta_l - q_{l,1}$$
$$q_{u,1} \leftarrow \langle \boldsymbol{v}, \boldsymbol{x}_u \rangle, \qquad q_{u,2} \leftarrow \|\boldsymbol{x}_u\|^2, \qquad U \leftarrow q_{u,2}/\Delta_u + q_{u,1}$$

7:      **if** $U \leq L$ **then**

8:          break           ▷ the for loop

9:      **end if**

10: **end for**

11: select vector $\boldsymbol{v}$, weight $t = 1/L$ and return $q_{l,1}, q_{l,2}, q_{u,1}, q_{u,2}$

---

## A.2 MODIFIED SINGLE SET SPARSIFICATION ALGORITHM

For implementing modified single-set sparsification, we replace the generic instruction at line 8 and computations at lines 9,10 in Algorithm 5 with the more detailed subroutine 6.

---

**Algorithm 5** Modified single-set sparsification algorithm

---

**Require:** $\{\boldsymbol{v}_i\}_{i=1}^m$ s.t. $\sum_{i=1}^n \boldsymbol{v}_i \boldsymbol{v}_i^\top = \boldsymbol{I}_n$, $N \geq n$,

**Ensure:** $\boldsymbol{A} = \sum_i t_i \boldsymbol{v}_i \boldsymbol{v}_i^\top$ s.t. $|\{i : t_i \neq 0\}| \leq r$, and $(1 - \sqrt{n/N})^2 \boldsymbol{I}_n \prec \boldsymbol{A} \prec (1 + \sqrt{n/N})^2 \boldsymbol{I}_n$

1: Let $\epsilon = \sqrt{n/N}$, $\kappa = (1+\epsilon)/(1-\epsilon)$, and

$$\delta_L = 1, \qquad \epsilon_L = \epsilon, \qquad \delta_U = \kappa, \qquad \epsilon_U = \epsilon/\kappa.$$

2: Initialize $\boldsymbol{A} = \boldsymbol{0}_{n \times n}$, $\boldsymbol{X} = [\,]$, $\boldsymbol{B} = [0]$,      ▷ matrices initialization

3: Initialize $\ell = -n/\epsilon_L$, $u = n/\epsilon_U$,      ▷ barriers initialization

4: Initialize $\phi_\ell = \epsilon_L$, $\phi_u = \epsilon_U$,      ▷ potentials initialization

5: let $\boldsymbol{V} = [\boldsymbol{v}_1, \ldots, \boldsymbol{v}_m] \in \mathbb{R}^{n \times m}$ and compute $\mathcal{E} = [\|\boldsymbol{v}_1\|^2, \ldots, \|\boldsymbol{v}_m\|^2]$,

6: Initialize $\boldsymbol{W} = [0, \ldots, 0]$,      ▷ cache initialization

7: **for** $k = 1, \ldots, N$ **do**

8:     select vector $\boldsymbol{v} \in \{\boldsymbol{v}_i\}$ and number $t > 0$ satisfying

$$U(\boldsymbol{v}, \delta_U, \boldsymbol{A}, u) \leq \frac{1}{t} \leq L(\boldsymbol{v}, \delta_L, \boldsymbol{A}, \ell).$$

9:     compute $q_{\ell,1} = \boldsymbol{v}^\top (\boldsymbol{A} - (\ell + \delta_L)\boldsymbol{I}_n)^{-1}\boldsymbol{v}$,   $q_{\ell,2} = \boldsymbol{v}^\top (\boldsymbol{A} - (\ell + \delta_L)\boldsymbol{I}_n)^{-2}\boldsymbol{v}$

10:    compute $q_{u,1} = \boldsymbol{v}^\top ((u + \delta_U)\boldsymbol{I}_n - \boldsymbol{A})^{-1}\boldsymbol{v}$,   $q_{u,2} = \boldsymbol{v}^\top ((u + \delta_U)\boldsymbol{I}_n - \boldsymbol{A})^{-2}\boldsymbol{v}$

11:    update

$$\Phi_\ell \leftarrow \Phi_{\ell+\delta_L} + \frac{q_{l,2}}{1/t - q_{l,1}}, \qquad \Phi_u \leftarrow \Phi_{u+\delta_u} - \frac{q_{u,2}}{1/t + q_{u,1}}$$

12:    compute $\mathbf{z} = [\langle \boldsymbol{v}, \boldsymbol{v}_1 \rangle, \ldots, \langle \boldsymbol{v}, \boldsymbol{v}_m \rangle]$,      ▷ $\mathbf{z} = \boldsymbol{V}^\top \boldsymbol{v}$

13:    update $\boldsymbol{X} \leftarrow [\boldsymbol{X}, \sqrt{t}\,\boldsymbol{v}]$, $\boldsymbol{B} \leftarrow \begin{pmatrix} \boldsymbol{B} & \sqrt{t}\,\boldsymbol{w} \\ \sqrt{t}\,\boldsymbol{w}^\top & t\,\xi \end{pmatrix}$, $\boldsymbol{W} \leftarrow \begin{pmatrix} \boldsymbol{W} \\ \sqrt{t}\,\mathbf{z} \end{pmatrix}$    ▷ matrices/cache

14:    update $\boldsymbol{A} \leftarrow \boldsymbol{A} + t\boldsymbol{v}\boldsymbol{v}^\top$, $\ell \leftarrow \ell + \delta_L$, $u \leftarrow u + \delta_U$

15: **end for**

16: multiply selected weights $t$ and $\boldsymbol{A}$ by $(1 - \sqrt{n/N})N^{-1}$

---

---

**Algorithm 6** selection of vector/weight $(\boldsymbol{v}, t)$ and associated vector $\boldsymbol{w}$ and squared norm $\xi$

---

1: compute $\boldsymbol{Z}_\ell = (\boldsymbol{B} - (\ell + \delta_L)\mathbf{I}_k)^{-1}$, and $\Phi_{\ell+\delta_L} = \text{Tr}(\boldsymbol{Z}_\ell) - (n-k)/(l + \delta_L)$,

2: compute $\boldsymbol{Z}_u = ((u + \delta_U)\mathbf{I}_k - \boldsymbol{B})^{-1}$, and $\Phi_{u+\delta_U} = \text{Tr}(\boldsymbol{Z}_u) + (n-k)/(u + \delta_U)$,

3: compute $\Delta_\ell = \Phi_\ell - \Phi_{\ell+\delta_L}$ and $\Delta_u = \Phi_{u+\delta_U} - \Phi_u$

4: consider variables $q_{l,1}, q_{l,2}, q_{u,1}, q_{u,2}, L, U$

5: **for** $i = 1$ to $m$ **do**

6:    let $\boldsymbol{v} = \boldsymbol{v}_i$, $\xi = \mathcal{E}_i$, and $\boldsymbol{w}$ be the $i$-th column of $\boldsymbol{W}$      ▷ $\xi = \|\boldsymbol{v}\|^2$ and $\boldsymbol{w} = \boldsymbol{X}^\top \boldsymbol{v}$

7:    compute $\mathbf{y}_l = \boldsymbol{Z}_l \boldsymbol{w}$, and $\mathbf{y}_u = \boldsymbol{Z}_u \boldsymbol{w}$

8:    compute

$$q_{l,1} \leftarrow \frac{\boldsymbol{w}^\top \mathbf{y}_l - \xi}{l + \delta_L}, \qquad q_{l,2} \leftarrow \frac{\|\mathbf{y}_l\|^2 - q_{l,1}}{l + \delta_L}, \qquad L \leftarrow q_{l,2}/\Delta_l - q_{l,1}$$

$$q_{u,1} \leftarrow \frac{\boldsymbol{w}^\top \mathbf{y}_u + \xi}{u + \delta_U}, \qquad q_{u,2} \leftarrow \frac{\|\mathbf{y}_u\|^2 + q_{l,1}}{u + \delta_U}, \qquad U \leftarrow q_{u,2}/\Delta_u + q_{u,1}$$

9:    **if** $U \leq L$ **then**

10:      break      ▷ the inner for loop

11:    **end if**

12: **end for**

13: select vector $\boldsymbol{v}$, weight $t = 1/L$ and return $\boldsymbol{w}, \xi$, and $q_{l,1}, q_{l,2}, q_{u,1}, q_{u,2}$

---

### A.3 RESTARTED/AGGREGATED SPARSIFICATION HEURISTIC

Let us consider $\epsilon = 1/\sqrt{d} \in ]0,1[$, the parameters $\epsilon_L, \epsilon_U, \delta_L, \delta_U$ as in 3 and execute single-set sparsification for $N > n$ iterations, as presented in the paper and without final normalization of $\boldsymbol{A}$ (line 12 in Algorithm 3 and line 16 in Algorithm 5) The output matrix $\boldsymbol{A}$ satisfies

$$(-n/\epsilon + N)\boldsymbol{I}_n \prec \boldsymbol{A} \prec \kappa(n/\epsilon + N)\boldsymbol{I}_n,$$

see equation 4. If we use $N = dn = n/\epsilon^2$ iterations, we obtain

$$nd(-\epsilon + 1)\boldsymbol{I}_n \prec \boldsymbol{A} \prec \kappa(\epsilon + 1)\boldsymbol{I}_n$$

If instead we execute sparsification for $n$ iterations and repeat this process $d$ times (with reshuffled $\{\boldsymbol{v}_i\}$ preferably), the individual output matrices $\boldsymbol{A}^{(j)}$ satisfy $\boldsymbol{0}_{n \times n} \prec \boldsymbol{A}^{(j)} \prec \kappa(n/\epsilon + n)\boldsymbol{I}_n$, hence the sum of output matrices $\boldsymbol{A}^{(j)}$ satisfies

$$\boldsymbol{0}_{n \times n} \preceq \left( \sum_{i=1}^{d} \boldsymbol{A}^{(i)} \right) /nd \prec \frac{1}{\epsilon}\kappa(\epsilon + 1)\boldsymbol{I}_n.$$

This last approach is faster, however provides worse estimate on the upper eigenvalue and no estimate on the lower eigenvalue. In practice, we can design heuristics that would compel lower eigenvalue of aggregated matrices $\boldsymbol{A}^{(i)}$ to quickly become nonzero and increase steadily.

We consider a very general outline for this Restarted/aggregated Algorithm.

---

**Algorithm 7** Fast restarted sparsification algorithm

---

**Require:** $\{\boldsymbol{v}_i\}_{i=1}^m$ s.t. $\sum_{i=1}^n \boldsymbol{v}_i\boldsymbol{v}_i^\top = \boldsymbol{I}_n$, $J \geq 1$, $0 < \epsilon < 1$
**Ensure:** $\boldsymbol{A} = \sum_i t_i \boldsymbol{v}_i \boldsymbol{v}_i^\top$ s.t. $|\{i : t_i \neq 0|\} \leq \sum_j N_j$,
 1: let $\kappa = (1 + \epsilon)/(1 - \epsilon)$ and define $\delta_L = 1, \epsilon_L = \epsilon, \delta_U = \kappa, \epsilon_U = \epsilon/\kappa$.
 2: Initialize $\boldsymbol{A} = \boldsymbol{0}_{n \times n}$,
 3: **for** $j = 1, \dots, J$ **do**
 4:     let $N_j$ be a number of rank-one matrices to be added
 5:     consider $\{\boldsymbol{v}_i\}$ reordered in a certain way
 6:     compute $\boldsymbol{W}_j =$ Algorithm 5 / Algorithm 3($\{\boldsymbol{v}_i\}, N_j, \delta_L, \epsilon_L, \delta_U, \epsilon_U$) without normalization (line 16/ line 12)
 7:     $\boldsymbol{A} \leftarrow \boldsymbol{A} + \boldsymbol{W}_j$
 8: **end for**

---

This heuristic has practical grounding. The main objective here is to improve complexity while emulating BSS algorithm. By Algorithm  Algorithm 5 / Algorithm 3, we mean the improved Algorithm 5 for up to $n$ iterations concluded by the plain Algorithm if needed. For the above algorithm to have better computational complexity than plain BSS, we need to have $N_j < n$ for all $j$.

The way we decide on the cardinality $N_j$ and how to reorder $\{\boldsymbol{v}_i\}$ at every iteration will affect greatly the performance of the algorithm. Whatever the strategy, we have a uniform bounding on largest eigenvalue of $\boldsymbol{A}$ at the end of iteration $j$, i.e. $\lambda_{max}(\boldsymbol{A}_j) \leq \lambda_{max}(\boldsymbol{A}_{j-1}) + \kappa(n/\epsilon + N_j)$. In particular, the output matrix $\boldsymbol{A}$ satisfies $\lambda_{max}(\boldsymbol{A}) \leq \kappa (Jn/\epsilon + N)$ where $N = \sum_{j=1}^{J} N_j$. This is to be compared with the upper bound $\kappa (n/\epsilon + N)$ insured by plain BSS Algorithm run for $N$ iteration. We have however no control over the smallest eigenvalue of $\boldsymbol{A}$. We note however that if $-n/\epsilon_L + N_j\delta_L \geq 0$ for at least one $j$, we are insured that the matrix $\boldsymbol{A}$ becomes definite positive during the algorithm.

In moderate as well as high dimensional setting ($n >> 1$), one can experiment with this algorithm for small values of $N_j$ in order to quickly produce sparse sums $\sum_j t_j \boldsymbol{v}_j \boldsymbol{v}_j$ and check afterward the well conditioning compared to $\kappa^2$ insured by plain BSS. We have experimented with fixed cardinality strategies. More precisely, we compare BSS run for $N \approx n/\epsilon^2$ iteration, with the heuristic parametrized with $c \in \{1, \dots, n\}$ and run for $N_j = c$ for $J \approx N/c$ iterations. The hyper-parameter $c$ is intended for fine tuning. As for reordering $\{\boldsymbol{v}_i\}$, we have experimented with

- **Strategy 1**: randomly reshuffle $\{\boldsymbol{v}_i\}$ at every iteration,

- **Strategy 2**: we let $\ell_j$ be a strict lower barrier to $\boldsymbol{A}$ at the $j$-th iteration, then reorder the vectors in $\{\boldsymbol{v}_i\}_i$ in decreasing order w.r.t. $\boldsymbol{v}^\top (\boldsymbol{A} - \ell_j \boldsymbol{I}_n)^{-1} \boldsymbol{v}$.

The lower barrier $\ell$ considered at the $k$-th iteration of plain BSS Algorithm is

$$\ell_0 + (k-1) = -n/\epsilon + (k-1),$$

which results from adding $(k-1)$ rank-one update to $\boldsymbol{A} = \boldsymbol{0}_{n \times n}$ and we are guaranteed this barrier become positive after $k \simeq n/\epsilon$ iteration. In the restarted/agregated algorithm, we have in general no guarantee $\boldsymbol{A}$ becomes definite positive, and we will consider such barriers only when negative. More precisely, for $j = 1, \ldots, J$, we consider the lower barrier $\ell_j = \min(-n/\epsilon + \sum_{i=1}^{j-1} N_i, -1 + \lfloor n/\epsilon \rfloor)$. We note that $-1 + \lfloor n/\epsilon \rfloor$ is the largest $-n/\epsilon + k$ located strictly below 0. In particular $\ell_j < 0$ for any $j = 1, \ldots, N$. For **strategy 2** we will consider this choice of $\ell_j$ (**strategy 2-1**) and also a more involved choice $\ell_j = \lambda_{\min}(\boldsymbol{A}_{j-1}) - \delta_L$ which entails computing the smallest eigenvalue of $\boldsymbol{A}$ at the beginning of the $j$-th iteration (**strategy 2-2**). We note that obviously the smallest eigenvalue of $\boldsymbol{A}$ is 0 for $\sum_{i=0}^{j-1} N_j < n$ rank-one update.

Below we revisit restarted Algorithm 7 with **strategy 2-2** which has given the best result. We simply run Algorithm 7 for $c$ iterations, then restart.

---

**Algorithm 8** Fast restarted sparsification algorithm

---

**Require:** $\{\boldsymbol{v}_i\}_{i=1}^m$ s.t. $\sum_{i=1}^n \boldsymbol{v}_i \boldsymbol{v}_i^\top = \boldsymbol{I}_n$, $N \geq 1$, $c \geq 1$, $0 < \epsilon < 1$
**Ensure:** $\boldsymbol{A} = \sum_i t_i \boldsymbol{v}_i \boldsymbol{v}_i^\top$ s.t. $|\{i : t_i \neq 0|\} \leq N$, and empirically $\lambda_{\max}(\boldsymbol{A})/\lambda_{\min}(\boldsymbol{A}) \leq \frac{(1+\epsilon)^2}{(1-\epsilon)^2}$
1: let $\kappa = \frac{1+\epsilon}{1-\epsilon}$ and define $\delta_L = 1$, $\epsilon_L = \epsilon$, $\delta_U = \kappa$, $\epsilon_U = \epsilon/\kappa$
2: Initialize $\boldsymbol{A} = \boldsymbol{0}$,
3: $\nu = 0$          ▷ Total number of added $t\boldsymbol{v}\boldsymbol{v}^\top$
4: **while** $\nu < N$ **do**
5:     let $\ell = \lambda_{\min}(\boldsymbol{A}) - \delta_L$ and compute $\boldsymbol{Z}_l = (\boldsymbol{A} - \ell \boldsymbol{I}_n)^{-1}$
6:     let $N_j$ be a number of rank-one matrices to be added      ▷ here the number is $c$
7:     $N_j \leftarrow \min(N_j, N - \nu)$         ▷ ensure $\nu = N$ at loop exit
8:     reorder $\{\boldsymbol{v}_i\}$ in decreasing order w.r.t. $\boldsymbol{v}^\top \boldsymbol{Z}_\ell \boldsymbol{v}$
9:     compute $\Delta_j =$ Algorithm 5$(\{\boldsymbol{v}_i\}, N_j, \delta_L, \epsilon_L, \delta_U, \epsilon_U)$ without normalization (line 16)
10:    update $\boldsymbol{A} \leftarrow \boldsymbol{A} + \Delta_j, \quad \nu \leftarrow \nu + N_j$
11: **end while**

---

Since every $\Delta_j$ satisfies $\boldsymbol{0} \prec \Delta_j \prec \kappa(n/\epsilon + N_j)\boldsymbol{I}_n$, then the final output matrix $\boldsymbol{A}$ satisfies $\boldsymbol{0} \prec \boldsymbol{A} \prec \kappa(Jn/\epsilon + N)$ where $J$ is the number of times the while loop was entered. We note that for values $\nu \leq n$, we have $\lambda_{\min}(\boldsymbol{A}) = 0$ and the computations of $\boldsymbol{Z}_l = (\boldsymbol{A} - \ell \boldsymbol{I}_n)^{-1}$ and of evaluations $\boldsymbol{v}^\top \boldsymbol{Z}_\ell \boldsymbol{v}$ can be performed via $\boldsymbol{B} = \boldsymbol{X}^\top \boldsymbol{X}$ assuming we have access to $\boldsymbol{X}$ in explained in Algorithm 5.

The time complexity of every iteration of this algorithm is $\mathcal{O}(n^3) + \mathcal{O}(mn^2 + m\log(m)) + T_j$ which corresponds to operations at lines 5 and 8, and $T_j$ the time complexity for computing $\Delta_j$. Assuming the numbers $N_j$ are fixed and are equal to $c$, we have $T_j = \mathcal{O}(mc\max(n, c^2))$ for all $j$ and $J \approx N/c$. Assuming $\log(m) \leq n^2$, the overall complexity is

$$\mathcal{O}\left(Nm\left(\frac{n^2}{c} + \max(n, c^2)\right)\right)$$

For example, for $c \approx n^{2/3}$, we have overall time complexity $\mathcal{O}\left(Nmn^{4/3}\right)$. For comparison, the overall time complexity of plain BSS algorithm is $\mathcal{O}\left(Nmn^2\right)$.

A.4   EXPERIMENTAL VALIDATION

The reported execution times were obtained on a personal laptop with a Dual-Core Intel i5. All algorithms were implemented in python numpy.

We let $n = 256$ and consider $G$ the complete weighted undirected $n$-vertex graph where vertices are numbered $s_1, \ldots, s_n$ and the weight on the edge connecting $s_i$ and $s_j$ is equal to $w_{i,j} = e^{-|i-j|/n}$.

We then consider the Laplacian $L_G = \sum_{1 \le i \ne j \le n} w_{i,j} (e_i - e_j)(e_i - e_j)^\top$ where $e_j$ are the unit vector in $\mathbb{R}^n$. We then consider the matrix $(\mathbf{1}\mathbf{1}^\top + L_G)$ which is symmetric definite positive and let $MM^\top$ be it Cholesky decomposition. We then consider the decomposition of identity

$$I_n = v_0 v_0^\top + \sum_{1 \le i \ne j \le n} v_{i,j} v_{i,j}^\top$$

where

$$v_0 := M^{-1}\mathbf{1}, \qquad v_{i,j} = \sqrt{w_{i,j}}\, M^{-1}(e_i - e_j), \qquad 1 \le i \ne j \le n.$$

The decomposition consists on a sum of $32641 = 1 + \frac{256 \times 255}{2}$ outer product. The vector are ordered as $v_{1,2}, \dots, v_{1,n}, v_{2,1}, \dots, v_{n,n-1}, v_0$

We let $d = 4$, $\epsilon = 1/\sqrt{d} = 1/2$ and consider parameters $\epsilon_L, \epsilon_U, \delta_L, \delta_U$ as in equation 3. In particular $\kappa^2 = 9$. We compare the execution times and condition numbers of matrices $A$ output by Algorithm **??** and Algorithm 7 with the discussed strategies.

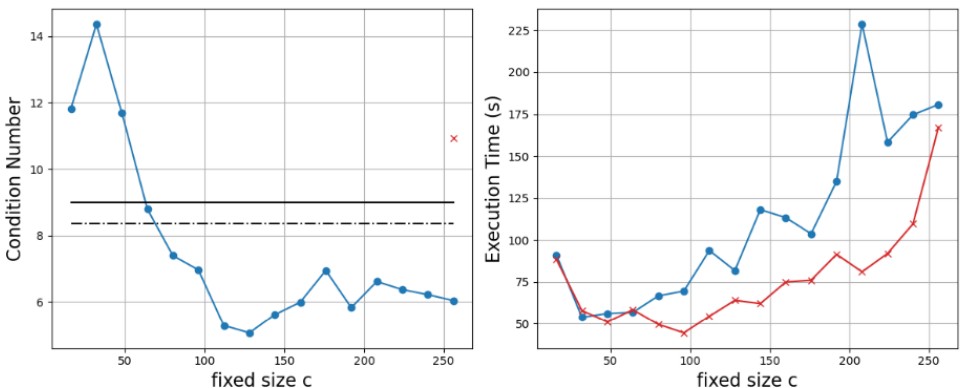

Figure 1: Comparison of condition number and execution time as a function of fixed size c.

The BSS algorithm run for $dn$ iterations took roughly 1200 seconds (20 minutes) and insure $\lambda_{\max}(A)/\lambda_{\min}(A) \approx 8.4 \le \kappa^2$. We also compare Algorithm 7 with **strategy 1** and **strategy 2-1** (in red) and **strategy 2-2** (in blue) for values $c \in \{16, 32, \dots, 256\}$ multiples of $\sqrt{n} = 16$. For **strategy 1**, the condition number is $\infty$ for all values of $c$ meaning the output matrix $A$ remains singular. This strategy run with $c = n/\epsilon = 2n$ took only 120 seconds but merely yields a condition number 100. **Strategy 2-1** is not reported in the figure, associated condition numbers are $[4604, 410, 427, 3346, 151, 128, 49, 105, 57, 35, 21, 38, 50, 43, 31, 11]$. **Strategy 2-2** is the most promising. For instance, the condition number for $c = 128$ is equal to 5. We note that the algorithm only took 60 seconds for this value.

Algorithm 7 performs very poorly with **strategy 1** and **strategy 2-1** but is very promising with **strategy 2-2**. We speculate the main reason is the following: the vectors $v_{i,j}$ have squared norms $\|v_{i,j}\|^2 = e^{-\frac{|i-j|}{2n}} (e_i - e_j)^\top (\mathbf{1}\mathbf{1}^\top + L_G)^{-1} (e_i - e_j)$ depend mostly in $|i - j|$. Any random shuffling or sorting w.r.t. to $v_{i,j}^\top (A - l_j I_n)^{-1} v_{i,j}$ for $l_j < 0$ will not create disparities among the $v_{i,j}$ hence not promoting those vectors that may push the smallest eigenvalue of $A$. It seems that **strategy 2-2** allow this. More experiments are needed for validating Algorithm 7 combined with intuitive heuristics such as **strategy 2-2**.

