# OpenReview forum: "improving time complexity of sparsification algorithms"
_ICLR.cc/2026/Conference — ICLR 2026 Conference Withdrawn Submission_

### Official Review · Reviewer_jW2A · 2025-10-26

**Soundness:** 3
**Presentation:** 3
**Contribution:** 2
**Rating:** 4
**Confidence:** 2

**Summary:**

This paper proposed spectral graph algorithms with improved time complexities. Technically, the proofs of this paper mainly follow from BSS-2009. They mainly utilize Sherman–Morrison-Woodbury formula to reduce computational costs of some matrices with low-rank structure. The proofs are sound and clearly written. The author(s) also proposed heuristic methods relying on restarting to improve numerical performance.

**Strengths:**

(1) The proofs of this paper are sound and easy to read.

(2) The dependency on $n$ in computational complexities are reduced to $k$ which is normally much smaller than $n$.

**Weaknesses:**

(1) The contributions to new design of spectral graph algorithms are marginal. The main framework is adopted from BSS-2009 with only matrix inversion part improved by utilizing low-rank structures and Sherman-Morrison-Woodbury formula. Its theoretical contribution is relatively marginal.

(2) The numerical experiments are minimal and could be substantially expanded to better demonstrate the performance of the algoithms proposed.

(3) The paper’s writing requires more careful proofreading. For example, there are ``??" on line 877.

(4) It would strengthen the paper to move some or all experimental results from the appendix into the main text.

**Questions:**

(1) Is there any theoretical guarantee for the proposed heuristic methods?

(2) Is there any analysis or empirical results on numerical stability?

(3) Is there any analysis on the tightness of the time complexity?

(4) Is it possible to improve the parallelism of the proposed algorithms?

---

### Official Review · Reviewer_DEju · 2025-10-29

**Soundness:** 2
**Presentation:** 2
**Contribution:** 2
**Rating:** 2
**Confidence:** 4

**Summary:**

This paper speeds up graph spectral and dual-set sparsification by rewriting each iteration's $n\times n$ matrix inverse into a smaller $k\times k$ (or Cholesky) computation and adding a simple restarted/aggregation heuristic, yielding faster, more stable implementations with empirical speedups.

**Strengths:**

The paper offers a clear, general speedup: at iteration $k$, it replaces costly $n\times n$ inverses with $k\times k$ (or Cholesky-lower) computations, cutting the dominant per-step cost. The linear-algebra justification (via Woodbury/Weinstein–Aronszajn) is clean and lets barrier-potential quantities be computed on the small matrix. The technique plugs into both BSS and dual-set frameworks, and in the dual-set case the paper provides an overall runtime bound. A small empirical study shows substantial wall-clock gains and improved conditioning.

**Weaknesses:**

The restarted/aggregated heuristic that drives the practical gains has no guarantee on the smallest eigenvalue, leaving a gap with classical BSS-style theory. The evaluation is limited to a single synthetic setup without comparisons to strong modern baselines. The benefit is front-loaded, most pronounced when $k\ll n$ and especially in dual-set regimes, and some steps (e.g., repeated $\lambda_{\min}$ computations) remain heavy; minor presentation glitches also reduce polish. Given this focus, the work seems less aligned with ICLR's typical scope and may be a better fit for venues centered on algorithmic engineering and numerical linear algebra.

**Questions:**

See weaknesses.

---

### Official Review · Reviewer_uyH5 · 2025-11-01

**Soundness:** 2
**Presentation:** 2
**Contribution:** 2
**Rating:** 2
**Confidence:** 3

**Summary:**

This paper studies the problem of deterministic spectral sparsification using algorithms like the Batson-Spielman-Srivastava (BSS) algorithm used for iteratively computing spectral sparsifiers of n-vertex graphs or the sum of rank 1 $n\times n$ matrices. A core contribution of this work is identifying the fact that for a low-rank PSD matrix $A$ there exists an equivalent low-dimensional representation that can be leveraged to speed-up any per-iteration inversion operation.

The main contribution is via Woodbury matrix identity which along with the Weinstein–Aronszajn identity gives a closed form representation of the inverse operation that are generally associated with the class of spectral sparsification algorithms that this paper is studying. Furthermore, this closed form operates in a smaller matrix space, which allows to achieve empirical speedup in practice. The resultant algorithm and computational improvements are well addressed in this paper.

**Strengths:**

I should pretext anything I mention by the fact that I have reviewed an older version of this paper.

- I believe as compared to the older version, the current version of the paper has improved and technically incorrect assumptions have been fixed. Although several issues remain. I talk about this in the weakness section.

- I really like the core idea of this paper -- it identifies a practical problem of computing inverses and finds a closed form solution via efficient matrix operations. This leads to improvements in runtime which is not shown in the main paper.

**Weaknesses:**

- I think the technical contribution of this work is pretty limited. An inverse operation is identified and improved via the Woodbury matrix identity. However, this identity has been well studied in the literature with applications in factor analysis and even matrix inverses. So clearly, I am unsure if this can be even considered a contribution.

- Although the main text is well written several issues remain. The core contribution does not appear until much later. I feel much of the space is wasted in prior work's exposition which limits the authors ability to include empirical results. Furthermore, there is only one empirical comparison. I think this is vastly insufficient. For a method that is claiming to achieve an empirical advantage, I feel this is insufficient. Other issues with writing like grammar, and incorrect links remain.

- The inverse operation assumes that the matrix is low-rank. Now one can clearly study low-rankness and approximation error due to that e Woodbury identity. However this is never studied in this paper. I think such a contribution could further add to the merits of the paper.

- Since the contribution is mostly empirical, one would expect an extensive empirical exposition. However this work presents only two plots in Figure 1 as only empirical comparison. It studies condition number and runtime of synthetic matrices vs the weight multiplier $c$, where $c = O(n^{2/3})$, the role of which is never really explained well. It would have been very helpful if clear discussion about variables were presented, or the methods were studied on real world scenarios. I understand that the compute power available to author is limited, but I also note that dual core i5 with 8GB of RAM can handle a wide variety of matrix inverses.

- Given the plethora of randomized and deterministic matrix sparsifiers, one would imagine a comparison to those algorithms is required for a better exposition.

**Questions:**

Please address the issues mentioned in the weaknesses section.

---

### Official Review · Reviewer_1wh5 · 2025-11-01

**Soundness:** 3
**Presentation:** 2
**Contribution:** 2
**Rating:** 4
**Confidence:** 3

**Summary:**

The paper studies the problem of improving computational complexity of a widely used spectral sparsification algorithm [BSS09]. The main observation is the inversion operation of (A-(l+delta_L) I_n) and ((u+delta_H) I_n -H) in each iteration with A=XX^T in R^{n * n} can be transformed to the inversion of involving B=X^TX in R^{k*k}. This leads to an improved time complexity, with the complexity provided in Line 853 in Appendix, O(Nm(n^2/c+max(n,c^2)). Empirical evaluations were conducted, but given in Appendix, to evaluate two strategies on reordering vectors {v_i}.

**Strengths:**

Strengths:
1. The BSS algorithm is a sparsification algorithm with great impacts in different problems in machine learning and applied math. Improvement on the time complexity thus has great potential in improving downstream applications. This is a main reason for getting relatively high point from me.
2. In the main paper, the authors are able to clearly describe the background of BSS sparsification, its algorithmic details, and the time improvements on its key steps. The good writing is true even though the paper organization should be enhanced, as will be discussed in the weaknesses.
3. The derivation of the computational improvements in the main text looks correct. The conversion from XX^T to X^TX successfully reduces the matrix inversion complexity.

**Weaknesses:**

Weaknesses:
1. While the title of the paper is on improving the time complexity of BSS09, the obtained improved complexity and the original complexity should be directly and explicitly given, best in the Introduction. The authors only briefly mention the complexity in Line 853, where the complexity is described in a blur way: when c is about n^{2/3}, the new complexity is O(Nmn^{4/3}) and the original complexity is O(Nmn^2).
2. The paper organization can be enhanced. First, the applications of BSS were extensively discussed with 1.5 pages. You could consider shorten them and move some to the Appendix. Second, there are still 1.5 pages left and it is natural to include the implementation of sparsification algorithms and the numerical study into the main text. It seems like the authors were in a rush to finalize the paper to get it submitted and did not get a chance to better organize and polish it.
3. The empirical study in the Appendix does not even include introduction on the graph data (?!). The main conclusions on the study and how they connect to the theory are not clear.

Comments:
- Some writing in the main text and the Appendix are sloppy, e.g., Lines 397, “which corresponds to Algorithm 2 improved complexity. in the case n2 > N and N ≤ n2”. There are quite a few typos as well, such as in Lines 897, 877, 326, and 291.
- The legends in Figure 1 are not provided.
- Source code for the empirical experiments is not provided.

**Questions:**

Please see the weaknesses above.

---

### Note · Authors · 2025-12-02

I have read and agree with the venue's withdrawal policy on behalf of myself and my co-authors.